# Socioeconomic disadvantage as a driver of non-urgent emergency department presentations: A retrospective data analysis

**Maria Unwin**[1,2]*, **Elaine Crisp**[1], **Jim Stankovich**[3☯], **Damhnat McCann**[1], **Leigh Kinsman**[4☯]

**1** School of Nursing, University of Tasmania, Launceston, Tasmania, Australia, **2** Emergency Department, Launceston General Hospital, Tasmania Health Service, Launceston, Tasmania, Australia, **3** Department of Neuroscience, Central Clinical School, Monash University, Melbourne, Victoria, Australia, **4** School of Nursing, University of Newcastle, Port Macquarie, New South Wales, Australia

☯ These authors contributed equally to this work.
* maria.unwin@utas.edu.au

**Data Availability Statement:** The data used in this analysis was provided by a third party (Tasmanian Department of Health and Human Services). In Australia this data is not publicly available, and

## Abstract

### Background

Globally, emergency departments (EDs) are struggling to meet the service demands of their local communities. Across Australia, EDs routinely collect data for every presentation which is used to determine the ability of EDs to meet key performance indicators. This data can also be used to provide an overall picture of service demand and has been used by health-care planners to identify local needs and inform service provision, thus, using ED presentations as a microcosm of the communities they serve. The aim of this study was to use ED presentation data to identify who, when and why people accessed a regional Australian ED with non-urgent conditions.

### Method and materials

A retrospective data analysis of routinely collected ED data was undertaken. This included data obtained over a seven-year period (July 2009 to June 2016) in comparison with the Australian Bureau of Statistics census data. Analysis included descriptive statistics to identify the profile of non-urgent attendees and linear regression to identify trends in ED usage.

### Results

This study revealed a consistently high demand for ED services by people with non-urgent conditions (54.1% of all presentations). People living in the most disadvantaged socioeconomic decile contributed to 36.8% of these non-urgent presentations while those under 25 years of age contributed to 41.1%. Diagnoses of mental health and behavioural issues and of non-specific symptoms significantly increased over the study period (p < 0.001) for both diagnostic groups.

permission has not been provided for the research team to make it publicly available. In order to access this data, the required ethics approvals must be obtained. Once this has been done the research team would be able to access this data by special request in writing to: The Secretary, Department of Health and Human Services, GPO Box 125, Hobart, Tasmania, 7001, Australia. All ABS data are available online. URLs are located in the Supporting Information.

**Funding:** Authors who received funding: MU, LK, EC Grant numbers awarded: One grant provided. Full name of funder: Clifford Craig Foundation URL: https://cliffordcraig.org.au/ No - the funder did not play a role in study design, data collection & analysis, decision to publish, or preparation of the manuscript.

**Competing interests:** No authors have competing interests.

## Conclusion

The over-representation by those from the most socioeconomically disadvantaged areas highlights an inequity in access to services. The over-representation by those younger in age indicates behavioural patterns based on age. These key issues faced by our local community and the disparity in current service provision will be used to inform future health policy and service planning.

## Introduction

Emergency departments have been described as a microcosm of the communities they serve, meaning that they encapsulate features of the wider community [1]. Challenges faced by emergency departments (EDs) can reflect deficits in community-based resources [2]. As increasing demands for ED services continue to be reported globally, it is timely and necessary to identify drivers of ED demand. In Australia, over 8.3 million people accessed ED services between July 2018 and June 2019 (335 per 1000 population), 48% of whom were triaged to the two least urgent triage categories [3]. The Australian Triage Scale (ATS) is a five-tiered triage system with ATS 4 and 5 being the least urgent categories, patients triaged to these categories are assessed as being safe to wait for one or two hours respectively [4]. For the purpose of this study, we refer to ATS 4 and 5 presentations as non-urgent. We are confident that this group of patients included some who could have had their needs met in a primary care setting.

International research investigating these least urgent presentations has identified drivers of ED demand such as: patients' perceived need for urgent attention [5–7]; age and gender [7–9]; access to alternative services [10–12], and socioeconomic position [6, 10, 13]. Identifying drivers specific to individual EDs can inform service planning [2]. Furthermore, a mismatch between the known causes of ED demand and solutions implemented was identified in a systematic review and highlights the need to develop interventions that address specific causes [14]. These external drivers contribute to the challenge for hospitals and health services in implementing successful and sustainable solutions.

Furthermore, our understanding of the demand for ED services is complicated by contextual differences. These differences challenge the successful implementation of solutions. Variation in demographic profiles, community healthcare needs and service availability influence how and when people access services, including the decision to present to an ED with a 'non-urgent' condition [15]. Socioeconomic position, for example, has been identified as having both a positive and negative correlation with populations accessing EDs. This correlation is observed to vary across contexts, with one study identifying greater representation by populations from mid-high socioeconomic areas [10] while others report greater representation from lower socioeconomic areas [6, 13]. Of the studies that reported age and gender, one found a higher incidence among middle aged females [7] while another found a higher incidence among young males [9]. These studies demonstrate the unique microcosm within EDs and provide an indication of healthcare needs within their respective wider communities.

Tasmania, Australia's smallest State, with a population of 517,000 [16], has the highest rate of non-urgent ED presentations, with 88,000 triaged as ATS 4 or 5 in 2018–19 [3]. This island State is separated into three geographic regions with governing health services in the North, Northwest and South all operating under the overarching jurisdiction of the Tasmanian Health Service [17]. The population of Northern Tasmania is older (median age 43 years compared to 38 years nationally) and more socioeconomically disadvantaged (median weekly income

$537.00-AU compared to $662.00-AU nationally) than other Australian regions [18], compounded by inequitable access to primary care services in regional and rural Tasmania [19]. There are considerable regional differences in the profile of ED patients across these three regions highlighting the importance of identifying trends and types of ED presentations to inform service planning [14]. These regional variations in population healthcare trends and the mismatch between identified causes and solutions to address ED demand highlight the importance of bringing together knowledge and understanding of the drivers for ED demand before implementation of sustainable solutions.

In research conducted in Northern Tasmania, 31% of patients who present to the ED with non-urgent conditions would have preferred to be managed by their general practitioner (GP) if they had been available [8]. The limited service options [19] in this community and the distance to alternative EDs (the nearest is a smaller rural facility located 90km from the study hospital) contribute to ED demand. Moreover, there are no private EDs or urgent care facilities in Northern Tasmania. Northern Tasmanian residents also have limited access to primary care services within the community once business hours have ended. Business-hours have been defined as between 0800 to 1800 Monday to Friday and 0800 to 1200 Saturdays; public holidays and all other times are considered after-hours [20]. These limited service options indicate potential challenges around timely access to alternative services.

Emergency departments are the 'canary in the coalmine' for health services and the communities they serve [1]. Demands for ED services are reflective of broader population healthcare needs [2] and are influenced by the availability of services within the community [10, 21]. The aim of this paper is to establish a profile of who, when and why ED services were accessed by people with non-urgent conditions. The objectives are to:

1. Develop a profile and identify trends in who is presenting and when;

2. Identify patterns in where people come from, including the socioeconomic position, and;

3. Identify trends in discharge diagnoses.

This paper forms part of a larger body of work using an explanatory sequential mixed method to gain a deeper understanding of factors contributing to the decision to present to an ED with non-urgent conditions and develop relevant and sustainable strategies for health service planning.

## Materials and methods

Retrospective analysis of routinely collected hospital data was undertaken for all presentations triaged as ATS 4 or 5 at a single regional ED, between 1 July 2009 and 30 June 2016. This consisted of data entered into the Emergency Department Information Systems (EDIS) by ED staff at time of the patient's presentation, or at the time of discharge. Variables used in this analysis included: date, day of week and time of arrival to the ED; gender; mode of arrival; suburb of residence; discharge diagnosis; discharge destination, and referral on discharge. The first six variables were entered into EDIS by the triage nurse or clerical staff at the patient's time of arrival. The latter three were added by the treating physician or nursing staff at the time of departure. Diagnoses are based on International Diagnostic Codes, revision 10, as outlined by the World Health Organisation [22]. It was beyond the scope of this project to review presentations across all triage categories.

Research ethics approval was granted by the Tasmanian Human Research Ethics Committee (H0016504). Deidentified data were provided by the Tasmanian Department of Health and

Human Services (DHHS). This data is not publicly available in Australia and permission was not provided for it to be made publicly available.

## Study setting & participants

This study was undertaken in a large regional hospital in Northern Tasmania with a total bed capacity of 300 and a 26 bed ED [23]. Serving as a referral centre for a population of 143,500 [24] dispersed across 20,000 square kilometres. Data used for this analysis was from July 2009 to June 2016, for ATS 4 and 5 presentations. The DHHS also provided the total count of all ED presentations by month across all triage categories so the proportion of ATS 4 and 5 could be calculated. Further explanation of the included study population is provided in Fig 1.

We have included all ATS 4 and 5 presentations who resided in the regional city (Launceston) and its surrounding suburbs. Excluding those from outside this region allowed us to develop a profile of who, when and why the local community choose to access ED services, thus focusing on local drives of ED demand. This area was defined by using statistical area (SA) codes allocated by the Australian Bureau of Statistics (ABS). The greater Launceston area has an SA3 code of 60201. All suburbs with this code were included in the study area and total population was 81,029 in 2016 [25]. Population growth in this region was just 2.5% between 2011 and 2016 compared to the national growth of 8.3% [25, 26].

Data relating to socioeconomic position was derived from ABS data. Five-yearly census data is used to calculate average values of various socioeconomic indexes across geographical areas, known as Socioeconomic Indexes for Areas (SEIFA). One of these is the Index of Relative Socioeconomic Disadvantage (IRSD), which is the preferred measure to use when investigating disadvantage or lack of disadvantage [27]. This index is based on national socioeconomic classification, and takes into account income and additional variables including unemployment, disability, sole-parent status, level of education, employment classification, etc. [27]. Each suburb is given a score based on these variables, the lower the score the greater the disadvantage. The ABS also aggregate suburbs into deciles, dividing Australia's population into ten evenly sized population groups. Ten percent of the Australian population fall into each decile with IRSD 1 being the 10% of those with greatest disadvantage and IRSD 10 being those with the greatest advantage. The histogram of IRSD scores has a long left-tail (at the end of greatest disadvantage), so the difference in disadvantage between decile 1 and decile 2 is larger than between other pairs of adjacent deciles [28]. The IRSD score and deciles were linked to ED data using the suburb of residence in order to determine socioeconomic position.

## Data analysis

Initial review of the data included all presentations to the regional ED triaged as ATS 4 or 5. The patient's suburb/town of residence was used to exclude attendees from outside this regional city. The decision to focus only on presentations from the local area was to gain greater insight and understanding of the local community and to limit outlying factors that may have influenced the decision by non-local attendees to present to the ED.

Descriptive statistics were calculated using SPSS [29] to summarise the profile of patients accessing the ED with non-urgent conditions throughout the seven-year study period. Linear regression was used to explore trends over time by mode of arrival, referral on departure, episode end status, time of arrival (in-hours versus after-hours) and International Classification of Diseases, version 10 (ICD-10) [22]. ABS national census data from 2011 and 2016 [25, 26] were used to calculate age-standardised presentation rates by suburb (age-standardised to the overall age distribution profile of the Launceston region in 2016), with linear interpolation used to estimate populations in years between 2011 and 2016. Linear regression, weighted by

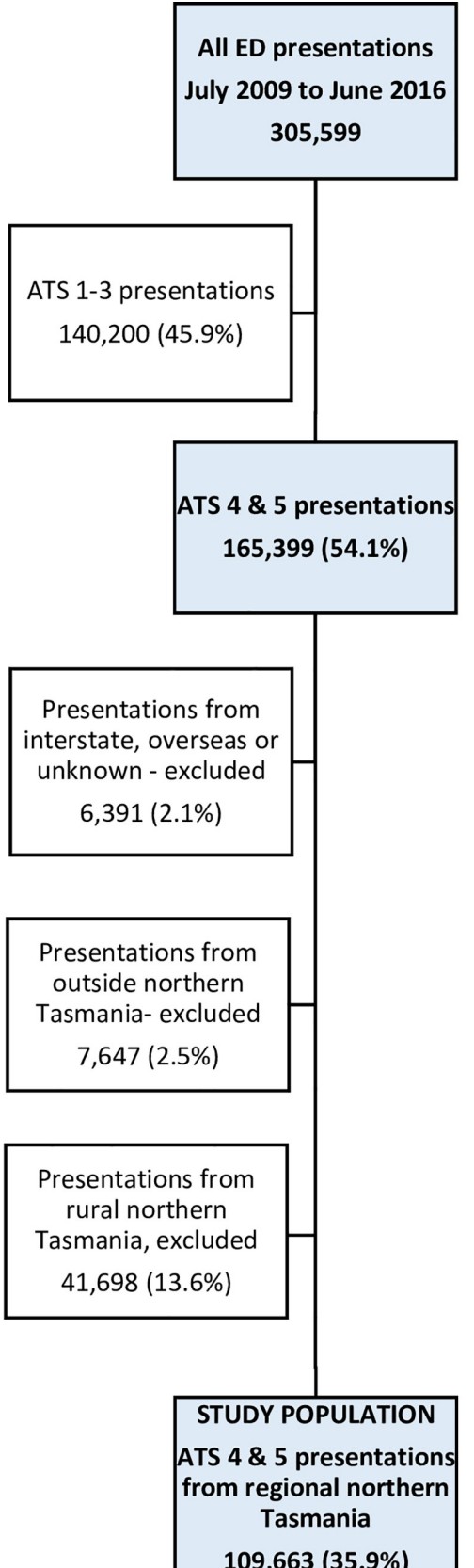

**Fig 1. Summary of ED presentation numbers, July 2009 to June 2016.**

2016 suburb populations, was used to fit a trend-line showing the association between age-standardized presentation rate and IRSD, with an outlier suburb excluded. RStudio [30] was used for regression analyses and plots.

## Results

Between 1 July 2009 to 30 June 2016, there were 305,599 ED presentations across all triage categories (ATS 1–5). Fig 1 provides a summary of how we determined the number (n = 109,633) included as the study population. Our objectives were to: describe the profile of ED attendees and trends over time through retrospective analyses of routinely collected hospital data; identify the usual place of residence and socioeconomic position of people attending the ED with non-urgent conditions, and to summarise the most frequent discharge diagnoses of the study population and trends over time.

### Profile and trends of people presenting with non-urgent conditions

The first objective was to develop a profile and identify trends in who is presenting and when. The number of non-urgent presentations to the ED revealed similar numbers between the first and last 12-month periods, July 2009 to June 2010 (n = 15,322) and July 2015 to June 2016 (n = 15,139). Over the seven-year study period the annual rate of non-urgent presentations among local residents varied between 186 to 205 per 1000 population. Fig 2A shows average daily rates by month of all non-urgent presentations. While there were short-term fluctuations in presentation numbers, regression analysis did not reveal any long-term linear trend in the number of presentations (p = 0.61). Over the seven-year study period non-urgent presentations by local residents ranged between 38 and 48 per day (Fig 2A).

Analysis of age identified that younger people were over-represented among non-urgent presentations. The median age of the study population was 29 years compared to a median age in this regional city of 39 years [31]. Table 1 provides a summary of presentation and population numbers aggregated by age. The age profile of the local population was recorded to remain stable between census periods, for example, those under 25 years of age continued to contribute to 31–33% of the local population between census periods.

Trends in mode of arrival revealed a consistency in the number and proportion of patients arriving by their own means (87%; Table 2). Analysis of presentation outcomes revealed a large proportion of patients either did not require any follow-up or were referred to their GP (74.7%; Table 2) and were discharged home from the ED (85.3%). For these two variables (arrival mode and presentation outcome), increases were observed in the number of patients with non-urgent conditions who: arrived by ambulance (average increase of 34 annually, p = 0.002); arrived with police (average increase of 56 annually, p<0.001), or who required admission to hospital (average increase of 56 annually, p<0.001).

Time of day and day of week are presented in Fig 2B with most non-urgent presentations occurring between 0800hrs and 1800hrs with peaks observed on Monday and Sunday mornings. Analysis of presentations occurring in-hours or after-hours revealed that 47.0% arrived in-hours with significant trends to in-hours and after-hours presentation numbers (2c and 2d). Average annual in-hours presentations fell at a rate of 78 per year (95% confidence intervals 18 to 140, p = 0.012). This was offset by a significant increase in after-hours presentations (rate of increase 108 annually, 95% confidence intervals 31 to 184, p = 0.006).

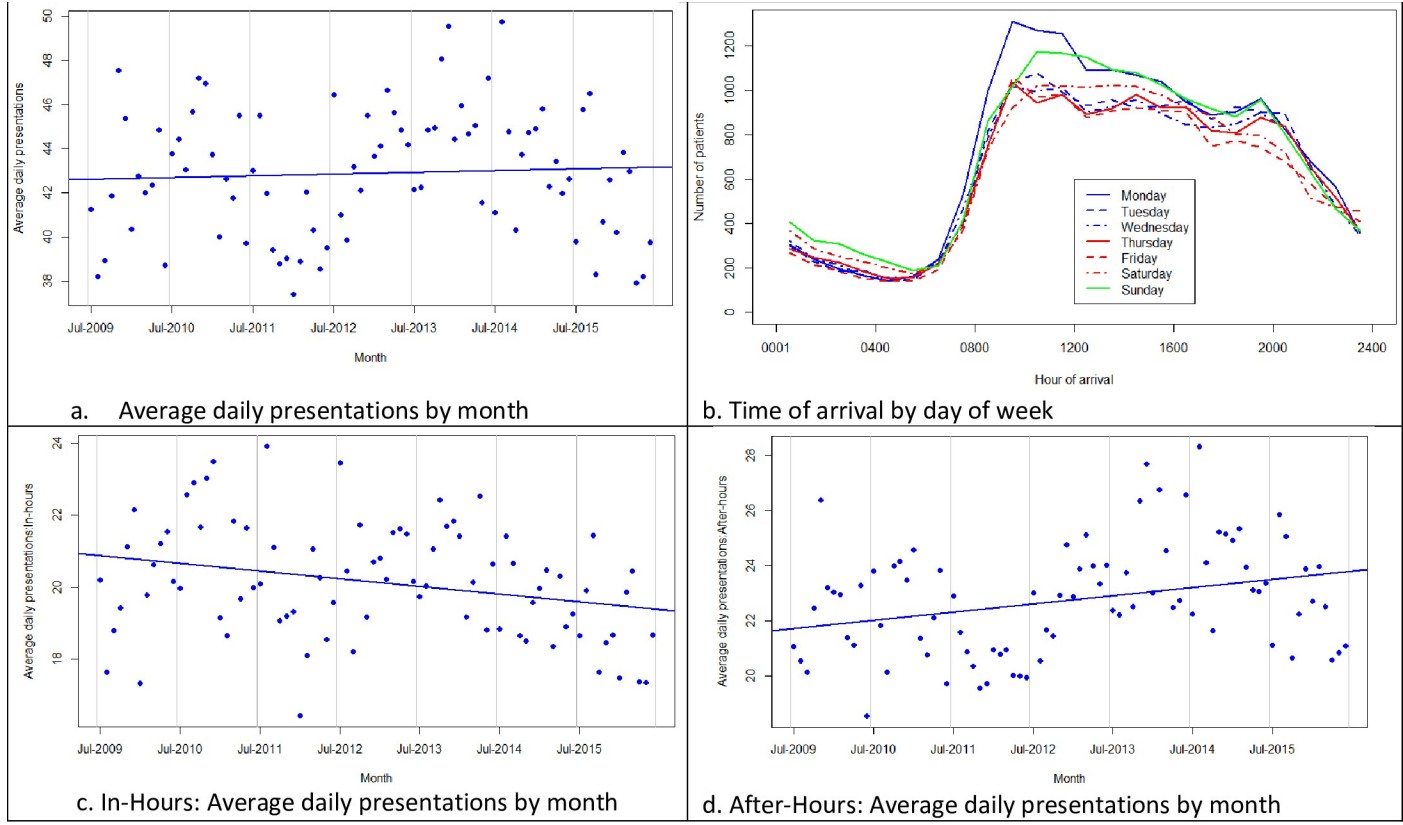

**Fig 2. Trends in presentation numbers and time of arrival, ATS 4 and 5, July 2009 –June 2016.** 2a. Average ATS 4 and 5 presentations by month, adjusted by days in month (p = 0.6). 2b. ATS 4 and 5 presentations, July 2009 to June 2016: time of day and day of week. 2c. Average in-hours ATS 4 and 5 presentations by month, adjusted by days in month (presentations 0800 to 1800 Monday to Friday and 0800 to 1200 Saturday). P-value for downward trend: 0.006. 2d. Average after-hours ATS 4 and 5 presentations by month, adjusted by days in month (presentations at times of week not included in Fig 2C, plus all presentations on public holidays). P-value for upward trend: < 0.001.

## Non-urgent ED attendees and socioeconomic levels

The second objective was to establish a profile based on the IRSD deciles according to the patient's suburb of residence. This age-standardised analysis revealed an over-representation by residents living in suburbs categorised as having the greatest socioeconomic disadvantage (IRSD decile 1; Table 1). Ten percent of the Australian population live in suburbs rated IRSD decile 1 compared to 26.4% of the Launceston population [25]. In this study, residents of IRSD decile 1 suburbs contributed to 36.8% of non-urgent ED presentations. Further analysis using the underlying IRSD score for each suburb revealed a strong negative correlation between IRSD score and the age standardised rate of ED attendance (Fig 3). Presentation rates for people with non-urgent conditions were 4.5 times higher from the most disadvantaged suburb compared to the most advantaged. Residents from the most advantaged suburb (IRSD score 1090) presented at a rate of 96 per 1000 population while residents from the most disadvantaged suburb (IRSD score 591) presented at a rate of 434 per 1000 population.

## Discharge diagnoses and trends over time

The number of presentations for the three most frequent overarching diagnostic groups are summarised in Table 3 along with the three most frequently recorded sub-diagnostic groups.

**Table 1. Profile of patients by gender, age and index for relative socioeconomic disadvantage IRSD) versus profile of local population, ATS 4 and 5, July 2009 to June 2016.**

| | *No.* | *% (n = 109 633)* | *% of local population (n = 81,029: ABS, 2016)** |
|---|---|---|---|
| *Gender* | | | |
| Male | 56 281 | 51.3 | 48.2 |
| Female | 53 293 | 48.6 | 51.8 |
| *Age (yrs)* | | | |
| **0–4** | **9 543** | **8.7** | **5.9** |
| 5–14 | 11 936 | 10.9 | 11.9 |
| **15–24** | **23 531** | **21.5** | **14.5** |
| 25–34 | 18 296 | 16.7 | 12.5 |
| 35–44 | 13 737 | 12.5 | 12.1 |
| 45–54 | 10 902 | 9.9 | 13.3 |
| 55–64 | 7 955 | 7.2 | 12.0 |
| 65–74 | 5 907 | 5.4 | 9.8 |
| 75–84 | 4 819 | 4.4 | 5.2 |
| 85+ | 3 037 | 2.8 | 2.5 |
| *IRSD by suburb (decile)*** | | | |
| **1 (greatest disadvantage)** | **40 379** | **36.8** | **26.4** |
| 2 | 5 058 | 4.6 | 1.2 |
| 3 | 9 993 | 9.1 | 9.5 |
| 4 | 20 098 | 13.1 | 22.5 |
| 5 | 8 218 | 7.5 | 3.6 |
| 6 | 11 576 | 10.6 | 17.8 |
| 7 | 1 828 | 1.7 | 7.6 |
| 8 | 4 562 | 3.9 | 5.6 |
| 9 | 1 080 | 1.6 | 1.6 |
| 10 (lowest disadvantage) | 413 | 0.4 | 0.8 |

*Australian Bureau of Statistics, 2016 Census Data Packs

** IRSD deciles divide 10% of nationwide population into each decile

Median age and results of linear regression analysis to determine trends in diagnostic groups are also reported in Table 3.

The most notable results from this analysis were the high proportion of discharge diagnoses falling into the ICD-10 code for injury. One third of non-urgent presentations were diagnosed with an 'injury, poisoning, certain other consequences of external causes', the most frequent sub-diagnostic groups were injuries to distal limbs or head. These patients were younger and there was no significant trend over the study period.

Significant increases in ED attendance were observed in two diagnostic groups, the first being 'symptoms, signs and abnormal clinical and laboratory findings, not elsewhere classified'. The proportion of patients diagnosed into this non-specific group increased from 6.6% in 2009–10 to 9.1% in 2015–16 ($p < 0.001$), the equivalent of 70 additional presentations per year.

Mental health conditions also increased significantly between 2009–16. These presentations increased from 1.8% of the study population to 3.1% ($p < 0.001$), a 73.1% increase in diagnoses relating to mental and behavioural disorders over seven years and equivalent to 31 additional presentations annually.

**Table 2. Summary and trends in ED presentations for mode of arrival and outcome of ED presentation, ATS 4 and 5, July 2009 –June 2016.**

| | No. | % (n = 109 663) | Trend: average annual change in presentations per year (95% confidence interval) | p-value for trend |
|---|---|---|---|---|
| *Mode of arrival* | | | | |
| Arrived by own means | 95 412 | 87.0 | −64 (−170, 41) | p = 0.2 |
| Ambulance | 12 350 | 11.3 | **34 (13, 55)** | **p = 0.002** |
| Police | 1 565 | 1.4 | **56 (44, 67)** | **p < 0.001** |
| Other | 336 | 0.3 | 2.2 (−0.5, 4.9) | p = 0.1 |
| *Referred to on departure* | | | | |
| GP or no further follow-up | 81 914 | 74.7 | 88 (−8, 184) | p = 0.07 |
| Emergency department | 7 370 | 6.7 | **−135 (−166, −103)** | **p < 0.001** |
| Outpatient department | 8 916 | 8.1 | −12 (−32, 8) | p = 0.2 |
| Community services | 3 010 | 2.7 | **10 (1, 20)** | **p = 0.03** |
| Hospital admission (same day) | 7 670 | 7.0 | **108 (93, 124)** | **p < 0.001** |
| Other hospital admission | 465 | 0.4 | **−4.7 (−8.3, 1.1)** | **p = 0.01** |
| Other | 318 | 0.3 | **−84 (−119, −49)** | **p < 0.001** |
| *Episode end status* | | | | |
| Discharged home | 93 567 | 85.3 | 1 (−114, 115) | p = 1.0 |
| Did not wait/Left at own risk | 8 571 | 7.8 | **−43 (−76, −9)** | **p = 0.01** |
| Admitted | 7 336 | 6.7 | **75 (56, 94)** | **p < 0.001** |
| Transferred | 161 | 0.1 | **−3.1 (−5.5, −0.8)** | **p = 0.01** |
| Other | 28 | 0.0 | 5.3 (−0.6, 11.3) | p = 0.08 |

## Discussion

This research aimed to identify who, when and why people accessed the ED with non-urgent conditions. In the analysis of seven-years' worth of routinely collected ED data, we discovered:

- No increase in total number of non-urgent presentations;

- A significant over-representation by residents from socioeconomically disadvantaged areas and those younger in age;

- Increasing proportion of after-hours presentations;

- Significant increases in presentations for mental health and non-specific symptoms.

### Consistent demand for ED services by patients with non-urgent conditions

The AIHW have consistently reported national increases in the number of annual ED presentations over the past five years [32], but an increase was not observed in the number of non-urgent presentations recorded to this ED during the study period. Monthly plots of presentation numbers demonstrate short-term fluctuations in ED usage for non-urgent conditions (Fig 2A and 2D), with presentation numbers between 186 to 205 per 1000 population per year. The simple linear regression we have performed does not adequately model fluctuations. Analysis of the fluctuations was beyond the scope of this publication but is part of an ongoing investigation by the research team.

A consistent demand for ED services by patients with non-urgent conditions has also been reported in research conducted in Northwest Tasmania where limited general practices services were identified as a driver [33]. Furthermore, international literature has identified links between the number of ED presentations and timely access to primary care services [10, 11, 21].

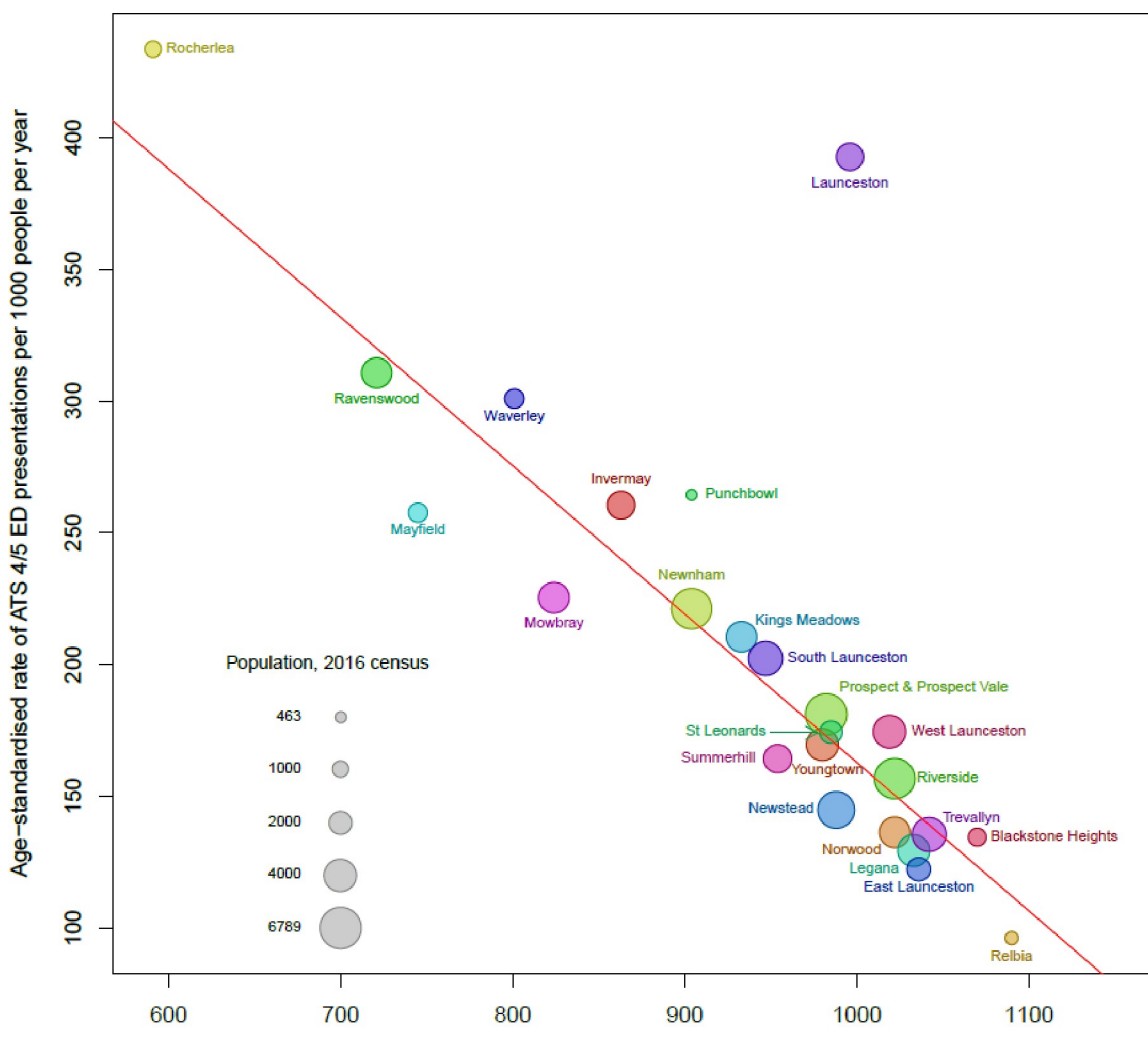

**Fig 3. Age standardised ED presentation rates for ATS 4 and 5.** Age standardised presentations per 1,000 (population), by suburb of residence and index for relative socioeconomic disadvantage (IRSD), July 2009 –June 2016.

Presentation numbers across day of week and time of day were observed to peak between 0900 and 1100hrs and decreasing throughout the day (Fig 2B). This indicates that a significant proportion of non-urgent presentations arrive during hours when other services are open. Tuesdays to Saturdays demonstrated similar presentation times and trends, however, peaks were observed on Sunday and Monday mornings. General practice services on a Sunday are minimal in this regional community leaving residents with the ED as the primary option. The peak on a Monday morning is likely to reflect those, who have waited for regular services to open on a Monday morning but been unable to secure an appointment, thus, resulting in an ED presentation. This again highlights the availability of alternative services at the time of need as a driver of non-urgent ED presentations and may be of interest to local service providers aiming to identify peak times and plan services and staffing based on demand.

**Table 3. Top three diagnostic groups and diagnostic groups with significant trends (based on international statistical classification of diseases and related health problems 10[th] Revision: ICD-10).** ATS 4 and 5, July 2009 to June 2016.

| Diagnosis, top ten ICD-10 In order of frequency Most frequent sub-diagnoses | No. presentations (% of sub-diagnostic group) | Proportion presentations (n = 109 663) (%) | Median age (IQR, years) | Trend over time |
|---|---|---|---|---|
| **XIX–Injury, poisoning, certain other consequences of external causes** | **36 567** | **33.3** | **25** (15–45) | No change |
| *Injuries to wrist and hand; head; ankle and foot* | *19 988 (54.7%)* | | | (p = 0.973) |
| **XXI–Factors influencing health status and contact with health services** | **14 980** | **13.7** | **33** (21–50) | No change |
| *Persons encountering health services for examination and investigation; in other circumstances; or for specific procedures and health care* | *14 443 (96.4%)* | | | (p = 0.156) |
| **XVIII–Symptoms, signs & abnormal clinical & laboratory findings, not elsewhere classified** | **8 442** | **7.7** | **34** (19–60) | **Significant increase** |
| *Symptoms and signs involving the digestive system and abdomen; general symptoms and signs; or involving the circulatory and respiratory systems* | *6 700 (79.4%)* | | | (p < 0.001) |
| **X–Diseases of respiratory system** | **7 024** | **6.4** | **22** (5–39) | **Significant decrease** |
| *Acute upper respiratory infections; chronic lower respiratory diseases; or influenza and pneumonia* | *6 340 (90.3%)* | | | (p = 0.002) |
| **V–Mental & behavioural disorders** *Neurotic, stress-related and somatoform* | **2 363** | **2.2** | **34** (23–48) | **Significant increase** |
| *disorders; mental and behavioural disorders due to psychoactive substance use; or Mood [affective] disorders* | *1 664 (70.4%)* | | | (p < 0.001) |

ICD-10 –International Classification of Diseases, version 10 [22]

IQR–Interquartile range

## Over-representation by those from lower socioeconomic suburbs and those younger in age

The correlation between IRSD and the number of non-urgent ED presentations per 1,000 head of population demonstrates a striking over-representation by people living in the most disadvantaged areas. The ED is located close to the central business district and is surrounded by suburbs with IRSD deciles between 3 and 7[25]. Furthermore, the suburb with the highest presentation numbers per 1,000 residents is the same distance from the ED as the suburb with the lowest presentation numbers, both being 11km from the ED. This shows that socioeconomic status is a stronger contributor to ED attendance than distance in our region. A higher proportion of non-urgent ED presentations by those living in close proximity has been previously reported [11, 34]; however, this was not the case in this study and highlights the contextual nature of how local populations access health services.

The only exception to the correlation between socioeconomic position and incidence of ED presentation (Fig 3) is the city centre. This appears to have occurred when the person providing the patient's details or staff member entering the data has listed the over-arching area of Launceston as the suburb of residence rather than the patient's actual suburb of residence. For example, it is not uncommon for residents from Launceston's lowest IRSD suburbs to list their suburb of residence as Launceston where it shares the same postcode as their actual suburb. These presentations were plotted in Fig 3 as they contribute to the overall number of presentations. However, the data from the city centre were excluded from the weighted regression analysis to fit a trend line due to the recording error.

Findings of over-representation among populations with greater socioeconomic disadvantage are varied across international literature. Some studies report similarly over-represented

presentations by disadvantaged communities [6, 13, 34] while a Canadian study found mid-high-income communities were over-represented [10]. Additionally, a study from the UK [12] reported that disadvantaged communities had lower ratios of GPs per 1,000 head of population. While it was outside the scope of this study to measure the number of GPs per 1,000 during the study period, it was observed that none of the larger practices with ready access to additional services such as pathology and radiology are located within the most disadvantaged areas of this local community. Furthermore, northern Tasmania was reported to have fewer full-time equivalent GPs in 2014, 70.3 per 1,000 population, versus 85.4 per 1,000 in southern Tasmania [35]. These findings highlight contextual differences in the ability of populations to access health services and demonstrates a disparity in the provision of healthcare services in the most socioeconomically disadvantaged areas of this community. Further supporting this finding, are two studies, one from the US focusing on paediatric presentations [36] and the other from New South Wales looking at all presentations (adult and paediatric) [37]. Both studies found that fewer GPs per 1000 population contributed to higher rates of non-urgent ED presentations.

Being younger in age was also a significant factor with a clear over-representation by those in the 0 to 4 and 15 to 24 age groups. These two groups were 1.5 times more likely to present with a non-urgent condition than the rest of the study population. This finding is consistent with international studies from the United States [6, 34], Canada [11], Switzerland[9, 38], the United Kingdom [13], and Australia [8, 39] all observing an over-representation in non-urgent presentations by younger populations. Consideration of why this over-representation is occurring may contribute to further understanding of the decision-making processes of young people and access to alternative services for this group.

It is likely that the over-representation of residents from socioeconomically disadvantage areas and by those younger in age is reflective of challenges faced by these populations in accessing the right service at the right time and located in the right place. This information will be of interest to future service planning.

## Increased non-urgent presentations after-hours

An increasing number of people arriving after-hours was also identified (Fig 2D). Most GP services in this community are available within normal business hours (0800 to 1800 weekdays and 0800–1200 Saturdays, excluding public holidays). Access to services is limited outside these times. The increase in demand for after-hours services is likely to reflect a lack in available services within the community at the time of need. Two other Tasmanian studies also found increases in after-hours presentations [17, 40] while another local study identified 31% of patients attending the ED would have preferred to be managed by their GP if they had been available at the time of need [8]. These findings further support the need for the right services to be available at the right time. As the third Tasmanian project to report a significant increase in the demand for after-hours services it is likely that further research exploring service demand and availability during these hours may assist in informing the provision of timely, patient-centred services and reduce ED demand.

## Increased presentations with non-urgent mental health diagnoses and with non-specific symptoms

The final objective was to identify prominent reasons for presentations through analysis of discharge diagnosis (Table 3) based on ICD-10 codes [22]. Unsurprisingly, presentations as a result of injury were the most common discharge diagnostic group with one third of all non-urgent presentations being as a result of 'injury, poisoning, certain other consequences of

external causes'. This is consistent with non-urgent presentations across Australia, the AIHW reporting that in 2017–18 [32], 32.7% of non-urgent ED presentations were allocated into this principle diagnostic group. Other studies have also found similar proportions for this diagnostic group [8, 39].

A significant increase was observed in diagnoses into the non-specific group of 'signs and symptoms or abnormal clinical findings not elsewhere classified'. This includes people who present to the ED for simple examination, investigation or observation, the proportion found in this study is reflective of nationwide trends for this principle diagnostic group [32]. The significant increase may be explained by international research which clearly identifies the patient's perceived need for urgent medical attention as a major theme when investigating reasons for accessing ED services with non-urgent conditions [6, 8, 38]. The continued high proportion of patients who were discharged home and did not require specialist follow-up in this study raises questions around health literacy, health anxiety and timely access to alternative services.

Diagnoses of 'mental and behavioural disorders' was the only other diagnostic group observed to increase significantly with an additional 30 people per year presenting to this regional ED. To the best of our knowledge, this patient group has not been identified as an increasing proportion of non-urgent ED presentations. In 2017–18 the AIHW recorded 2.6% of ATS 4 and 5 presentations resulting in a mental health or behavioural diagnosis, for the same period this regional ED observed 3.1% [32]. While these are similar proportions to national figures, we were able to identify a concerning increase of 73.1% between 2009–10 and 2015–16 in our regional ED. Limitations in AIHW reporting meant we were not able to compare this increase with earlier national numbers. A patient triaged as an ATS 4 or 5 with a mental health presentation must demonstrate the ability to provide a clear history without signs of restlessness or aggression [4].

It is not known what has caused this dramatic increase in mental and behaviour diagnoses within the local region. However, if the ED provides an indication of people's healthcare needs and the level of access to services within the community, this increase must be a warning to local service providers. Mental health was identified as the predominant concern for young people in a 2018 national survey of over 28,000 participants aged 15 to 19 years [41]. This report identified for the first time in 17 years that the top concern for youth was mental health. This growing concern among young people and the increasing presentation numbers within this regional community provide policy makers and service providers with a clear local need.

## Limitations

This longitudinal observational study was reliant on routinely collected hospital data; efforts were made to review data for possible discrepancies. The findings are largely reliant upon the quality of data collected at the time of the patients' presentation. Population and socioeconomic position data were based upon ABS data collected in 2011 and 2016 with changes occurring across this time period, to allow for these changes we presumed a direct linear relationship between the two data collection periods. This may not reflect true numbers but provided the closest solution to changes available between these two time periods.

Data provided by the DHHS were for ATS 4 and 5 presentations only, therefore it was not possible to compare presentation trends across all triage categories. This broader analysis was beyond the scope of this project and highlights an area for future enquiry.

## Conclusion

The ED is a 'canary in the coalmine' for the greater health service and community. The over-representation of population groups and increases in demand provide clear indicators of the

healthcare needs of members of the local community. Patients presenting to this regional ED with non-urgent conditions were younger than the local demographic profile and up to four times more likely to live in the most disadvantaged communities, raising the question of service accessibility and availability in areas of need. In addition, patients are increasingly presenting with non-specific symptoms and with mental health and behavioural issues. These findings will be of use to policy-makers in planning for enhanced primary care service for the young and for people with mental health issues from our most disadvantaged communities.

## Supporting information

**S1 File.**
(DOCX)

## Acknowledgments

The authors would like to thank the Tasmanian Health Service and the Department of Health and Human Services for their ongoing support of this study.

## Author Contributions

**Conceptualization:** Maria Unwin.

**Formal analysis:** Maria Unwin, Jim Stankovich, Leigh Kinsman.

**Funding acquisition:** Maria Unwin, Elaine Crisp, Leigh Kinsman.

**Project administration:** Maria Unwin.

**Software:** Maria Unwin, Jim Stankovich.

**Supervision:** Elaine Crisp, Damhnat McCann, Leigh Kinsman.

**Validation:** Maria Unwin, Elaine Crisp, Jim Stankovich, Damhnat McCann, Leigh Kinsman.

**Writing – original draft:** Maria Unwin.

**Writing – review & editing:** Maria Unwin, Elaine Crisp, Jim Stankovich, Damhnat McCann, Leigh Kinsman.

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
