## [Decision Letter · Decision Letter 0]

30 Jan 2020

PONE-D-19-35242

Socioeconomic disadvantage as a driver of non-urgent emergency department presentations: A retrospective data analysis

PLOS ONE

Dear Ms Unwin,

Thank you for submitting your manuscript to PLOS ONE. After careful consideration, we feel that it has merit but does not fully meet PLOS ONE’s publication criteria as it currently stands. Therefore, we invite you to submit a revised version of the manuscript that addresses the points raised during the review process.

Please pay particular attention to the technical issues raised by Reviewer 1. Also, please consider the comments of Reviewer 2 about limitations of this study.

We would appreciate receiving your revised manuscript by Mar 15 2020 11:59PM. To enhance the reproducibility of your results, we recommend that if applicable you deposit your laboratory protocols in protocols.io, where a protocol can be assigned its own identifier (DOI) such that it can be cited independently in the future. For instructions see: http://journals.plos.org/plosone/s/submission-guidelines#loc-laboratory-protocols

We look forward to receiving your revised manuscript.

Kind regards,

M Barton Laws

Academic Editor

PLOS ONE

Journal Requirements:

2. In ethics statement in the manuscript and in the online submission form, please provide additional information about the patient records used in your retrospective study. Specifically, please ensure that you have discussed whether all data were fully anonymized before you accessed them and/or whether the IRB or ethics committee waived the requirement for informed consent. If patients provided informed written consent to have data from their medical records used in research, please include this information.

3. In your Methods section, please provide additional information about the methods used in your study. Please ensure you have provided sufficient details to replicate the analyses such as a description of any inclusion/exclusion criteria that were applied to participant inclusion (for example, please specify the definition of "regional northern Tasmania", and a definition of the variables used (for example, please describe how "discharge diagnosis" was coded).

4. Please include your tables as part of your main manuscript and remove the individual files. Please note that supplementary tables (should remain/ be uploaded) as separate "supporting information" files.

Additional Editor Comments:

Please pay particular attention to the technical issues raised by Reviewer 1. Also, please consider the comments of Reviewer 2 about limitations of this study.

Reviewers' comments:

Reviewer's Responses to Questions

**Comments to the Author**

1. Is the manuscript technically sound, and do the data support the conclusions?

Reviewer #1: Partly

Reviewer #2: Yes

2. Has the statistical analysis been performed appropriately and rigorously? 

Reviewer #1: No

Reviewer #2: Yes

3. Have the authors made all data underlying the findings in their manuscript fully available?

Reviewer #1: No

Reviewer #2: No

4. Is the manuscript presented in an intelligible fashion and written in standard English?

Reviewer #1: Yes

Reviewer #2: Yes

5. Review Comments to the Author

Reviewer #1: This paper examines the characteristics of patients presenting in an ED for non-urgent reasons using routinely collected hospital data from a large regional hospital in northern Tasmania over the period July 1, 2009 to June 30, 2016. Data was provided by the Tasmanian Department of Health and Human Services. ED utilization and “crowding” is an important topic as it affects hospital emergency departments globally. A better understanding of the populations that are utilizing the ED for non-urgent reasons could help to target strategies to redirect patients to more appropriate health care resources. The authors find an over-representation of patients from the lower socioeconomic suburbs and younger in age. They also find growth in the volume of patients presenting with non-urgent ‘mental and behavioral disorders.’ The paper would be strengthened with a more clearly articulated methods section among other recommended improvements.

1. Materials and Methods – several questions arose in reading through the methods.

a. Page 6, lines 114-119. The authors describe bulk-billing and the fact that Tasmania has the lowest rate of bulk-billing. It seems that bulk-billing may have a large affect on an individual’s decision to seek care in an ED or a GP’s office. Yet, this is the only mention of the practice. It seems the author’s may want to include this in the discussion. It is also unclear if the authors have any information regarding bulk-billing within the various suburbs as that would be good information to include in the analysis.

b. Page 7, lines 149-153. It is unclear how the authors are using the IRSD deciles. Is it Australia’s total population that is broken into deciles, and then the suburbs within the study region are classified based on the national classification? If so, it would be helpful to have some information provided in a table related to the suburbs in the region or at least some more information on characteristics of each decile. For instance, what does it mean to be in decile 1 versus decile 2?

c. Methods – the presentation of methods is somewhat sporadic. It would be helpful to better organize the methods section. For instance, perhaps describe the variable construction first and then describe the analytic method. Right now the two seem intertwined and it makes the section unclear.

d. Methods – it might be helpful to compare the non-urgent to the urgent presentations in addition to the overall population.

2. Discussion – several clarifications are suggested in the Discussion

a. Page 11, line 233 – to the point that there is no increase in the total number of non-urgent presentations, it would be helpful to know how the population is growing (or not). For instance, if the ED volume is flat, but the population is growing, then the overall utilization is declining. Perhaps it would be helpful to report this as a utilization per 1,000 population or similar.

b. Page 11, line 243 – the authors state, “the erratic variation in presentation numbers points to external factors such as, the availability of primary care services…” However, Figure 2b shows a surge in ED volume around 0800, which, during the week, is when primary care services are available. Perhaps it correlates to bulk-billing. Please clarify.

c. Page 12, line 260. The authors state that they are certain the data from the city centre is clerical error. If so, why include in the study or why not at least run a sensitivity analysis with that suburb excluded.

3. Page 10, line 208. It is unclear where the “disadvantaged suburbs were up to four times more likely to present…” comes from. It is not clear in the data where that number is listed or if it was author calculation. This occurs in other areas of the manuscript as well. Please ensure that the results are clearly tied to the analysis.

4. Figure 2b appears to be fairly striking. The profile of presentation to the ED by time and day is consistent, regardless of access to other services being open. There seems to be a lot more here that is not really discussed in detail.

5. Figure 2a (and 2d to some extent) appears to have some sort of non-linear trend. Please clarify.

6. Table 2. For the trend, it might be helpful to have an actual table of regression results rather than just stating “increased,” “no change,” or “decrease.”

7. Table 3. There is a lot to digest in Table 3. Perhaps there is a better way to present this information. Also, are the subcategories just the top 3 within each?

Reviewer #2: The goal of this paper is to describe the population accessing the emergency department (ED) at a large regional hospital in Tasmania, including the demographics of the population and changes in utilization over time. The authors use routinely collected hospital data (on all admissions with an ATS value of 4 or 5, indicating the least urgent categories of visits, which the authors state indicate the patient could have been treated in a setting other than the ED. The authors find overrepresentation of patients from areas with a low socioeconomic status and distinct patterns by age. The paper is well written and the objectives clearly stated throughout. The introduction could be streamlined to make a better argument for the need for this analysis.

Introduction:

• The introduction sets the reader up to think that this will be a comparison of “local” environments and how that contributes to the demand for services, and in particular non-urgent services. But the paper only onlooks at one community (an important one), and so it’s so much a point about how the local environment matters so much as a depiction of what is going on in one particular local community. I think the introduction could be strengthened greatly by focusing more on why THIS community is important to study, as opposed to why local communities are important more broadly.

Methods

• The authors state that they used routinely collected hospital data, but as a reader it is not clear what this means. There are references to discharge diagnoses, but a more comprehensive description of the type of data used would be helpful.

• For context, it would be helpful to know how the changes in the presentation for a non-urgent condition compare to the overall, or to the presentations for urgent conditions. For instance, if the patterns in non-urgent conditions were simply representative of the overall patterns, then it wouldn’t necessarily suggest something specific about the need for non-urgent care and would likely change the authors interpretation.

• The authors exclude people from outside of the local community, but what about the potential for people within the community to seek care elsewhere? How much of the patterns you see are due to self selection of the population into this particular hospital?

Results

• The results are nicely written and clear. As are the figures and tables.

Discussion

• The authors state that “the erratic variation in presentation numbers points to external factors such as, the availability of primary care services, contributing to ED attendance for non-urgent conditions among the local population.” Yet, the authors do not point to any evidence in their own analysis that this is the case and in the introduction say that prior research has suggested that the literature on this topic is fairly mixed.

•

6. PLOS authors have the option to publish the peer review history of their article (what does this mean?). If published, this will include your full peer review and any attached files.

Reviewer #1: No

Reviewer #2: No

---

## [Author Response · Author response to Decision Letter 0]

20 Mar 2020

Response to Reviewers

REVIEWERS GENERAL COMMENTS: 

1. Is the manuscript technically sound, and do the data support the conclusions? The manuscript must describe a technically sound piece of scientific research with data that supports the conclusions. Experiments must have been conducted rigorously, with appropriate controls, replication, and sample sizes. The conclusions must be drawn appropriately based on the data presented. Reviewer #1: Partly Reviewer #2: Yes 

We hope the amended article addresses reviewer #1’s concerns.

2. Has the statistical analysis been performed appropriately and rigorously? Reviewer #1: No Reviewer #2: Yes 

We hope that, by addressing Reviewer #1’s specific concerns as outlined below, the reviewer is convinced that the statistical analysis has been performed appropriately and rigorously.

3. Have the authors made all data underlying the findings in their manuscript fully available? The PLOS Data policy requires authors to make all data underlying the findings described in their manuscript fully available without restriction, with rare exception (please refer to the Data Availability Statement in the manuscript PDF file). The data should be provided as part of the manuscript or its supporting information, or deposited to a public repository. For example, in addition to summary statistics, the data points behind means, medians and variance measures should be available. If there are restrictions on publicly sharing data— e.g. participant privacy or use of data from a third party—those must be specified. Reviewer #1: No Reviewer #2: No 

The data used in this analysis was provided by a third party (Tasmanian Department of Health and Human Services). In Australia this data is not publicly available, and permission has not been provided for the research team to make it publicly available.

In order to access this data, the required ethics approvals must be obtained. Once this has been done the research team would be able to access this data by special request in writing to: The Secretary, Department of Health and Human Services, GPO Box 125, Hobart, Tasmania, 7001, Australia

4. Is the manuscript presented in an intelligible fashion and written in standard English? PLOS ONE does not copyedit accepted manuscripts, so the language in submitted articles must be clear, correct, and unambiguous. Any typographical or grammatical errors should be corrected at revision, so please note any specific errors here. Reviewer #1: Yes Reviewer #2: Yes 

REVIEWER # 1: AUTHOR RESPONSE:

Reviewer #1: This paper examines the characteristics of patients presenting in an ED for non-urgent reasons using routinely collected hospital data from a large regional hospital in northern Tasmania over the period July 1, 2009 to June 30, 2016. Data was provided by the Tasmanian Department of Health and Human Services. ED utilization and “crowding” is an important topic as it affects hospital emergency departments globally. A better understanding of the populations that are utilizing the ED for non-urgent reasons could help to target strategies to redirect patients to more appropriate health care resources. The authors find an over-representation of patients from the lower socioeconomic suburbs and younger in age. They also find growth in the volume of patients presenting with nonurgent ‘mental and behavioral disorders.’ The paper would be strengthened with a more clearly articulated methods section among other recommended improvements. 

The methods section has been revised as suggested by reviewer #1. 

(Pg 6 – 8)

1. Materials and Methods – several questions arose in reading through the methods. 

a. Page 6, lines 114-119. The authors describe bulk-billing and the fact that Tasmania has the lowest rate of bulk-billing. It seems that bulk-billing may have a large affect on an individual’s decision to seek care in an ED or a GP’s office. Yet, this is the only mention of the practice. It seems the author’s may want to include this in the discussion. It is also unclear if the authors have any information regarding bulk-billing within the various suburbs as that would be good information to include in the analysis. 

Thank-you for pointing this out. Bulk billing discussion removed. This study did not investigate cost or bulk-billing practices so decision by research team to remove comments from this paper. This is part of an ongoing investigation by the research team.

b. Page 7, lines 149-153. It is unclear how the authors are using the IRSD deciles. Is it Australia’s total population that is broken into deciles, and then the suburbs within the study region are classified based on the national classification? If so, it would be helpful to have some information provided in a table related to the suburbs in the region or at least some more information on characteristics of each decile. For instance, what does it mean to be in decile 1 versus decile 2? 

Discussion of IRSD deciles amended and moved to ‘study setting and participants’ (pg 7 line 141 - pg – 156)

c. Methods – the presentation of methods is somewhat sporadic. It would be helpful to better organize the methods section. For instance, perhaps describe the variable construction first and then describe the analytic method. Right now the two seem intertwined and it makes the section unclear. 

Methods reformatted to separate variable construction and analytic method. (pgs 6 – 8)

d. Methods – it might be helpful to compare the non-urgent to the urgent presentations in addition to the overall population. 

Addressed in limitations, data were only provided for ATS 4 and 5 presentations (pg 20, line 414-416

2. Discussion – several clarifications are suggested in the Discussion 

a. Page 11line 233 – to the point that there is no increase in the total number of nonurgent presentations, it would be helpful to know how the population is growing (or not). For instance, if the ED volume is flat, but the population is growing, then the overall utilization is declining. Perhaps it would be helpful to report this as a utilization per 1,000 population or similar. 

Paragraph amended to include requested information. (pg 15, line 280-285)

b. Page 11, line 243 – the authors state, “the erratic variation in presentation numbers points to external factors such as, the availability of primary care services…” However, Figure 2b shows a surge in ED volume around 0800, which, during the week, is when primary care services are available. Perhaps it correlates to bulk-billing. Please clarify. 

Have clarified discussion of these figures. (pg 15, 280-285)

c. Page 12, line 260. The authors state that they are certain the data from the city centre is clerical error. If so, why include in the study or why not at least run a sensitivity analysis with that suburb excluded. 

c. We think it is important to plot the data from the city centre in Figure 3, as these are actual ED presentations and contribute substantially to the overall number of presentations. The data from the city centre has been excluded in the weighted regression analysis to fit a trend line. We apologize for failing to mention this in the original manuscript. This has now been added (pg 18, line 314-322)

3. Page 10, line 208. It is unclear where the “disadvantaged suburbs were up to four times more likely to present…” comes from. It is not clear in the data where that number is listed or if it was author calculation. This occurs in other areas of the manuscript as well. Please ensure that the results are clearly tied to the analysis. 

Further explanation provided (pg 13, line 228-239)

4. Figure 2b appears to be fairly striking. The profile of presentation to the ED by time and day is consistent, regardless of access to other services being open. There seems to be a lot more here that is not really discussed in detail. 

Further explanation provided (pg 17, line 291-301) 

5. Figure 2a (and 2d to some extent) appears to have some sort of non-linear trend. Please clarify. 

Explanation provided (pg 14, line 278 - pg 15, line 285)

6. Table 2. For the trend, it might be helpful to have an actual table of regression results rather than just stating “increased,” “no change,” or “decrease.” 

Table 2 amended; regression results now included.

7. Table 3. There is a lot to digest in Table 3. Perhaps there is a better way to present this information. Also, are the subcategories just the top 3 within each? 

Table 3 simplified, yes the sub-categories are the top 3 within each.

REVIEWER # 2: AUTHOR RESPONSE

Reviewer #2: The goal of this paper is to describe the population accessing the emergency department (ED) at a large regional hospital in Tasmania, including the demographics of the population and changes in utilization over time. The authors use routinely collected hospital data (on all admissions with an ATS value of 4 or 5, indicating the least urgent categories of visits, which the authors state indicate the patient could have been treated in a setting other than the ED. The authors find overrepresentation of patients from areas with a low socioeconomic status and distinct patterns by age. The paper is well written and the objectives clearly stated throughout. The introduction could be streamlined to make a better argument for the need for this analysis. 

Have re-structured introduction and moved Tasmanian background from methods to introduction. 

Introduction: 

• The introduction sets the reader up to think that this will be a comparison of “local” environments and how that contributes to the demand for services, and in particular nonurgent services. But the paper only onlooks at one community (an important one), and so it’s so much a point about how the local environment matters so much as a depiction of what is going on in one particular local community. I think the introduction could be strengthened greatly by focusing more on why THIS community is important to study, as opposed to why local communities are important more broadly. 

As above

Methods 

• The authors state that they used routinely collected hospital data, but as a reader it is not clear what this means. There are references to discharge diagnoses, but a more comprehensive description of the type of data used would be helpful. 

Clarified (pg 6, line 110 – 120)

• For context, it would be helpful to know how the changes in the presentation for a nonurgent condition compare to the overall, or to the presentations for urgent conditions. For instance, if the patterns in non-urgent conditions were simply representative of the overall patterns, then it wouldn’t necessarily suggest something specific about the need for nonurgent care and would likely change the authors interpretation. 

Sentence added to body of paper and to limitations of study (pg 6, line 119-120 & pg 22, line 414-416)

• The authors exclude people from outside of the local community, but what about the potential for people within the community to seek care elsewhere? How much of the patterns you see are due to self selection of the population into this particular hospital? 

Explanation provided (pg 7, line 136-140).

Results 

• The results are nicely written and clear. As are the figures and tables. 

Discussion 

• The authors state that “the erratic variation in presentation numbers points to external factors such as, the availability of primary care services, contributing to ED attendance for non-urgent conditions among the local population.” Yet, the authors do not point to any evidence in their own analysis that this is the case and in the introduction say that prior research has suggested that the literature on this topic is fairly mixed. 

Clarified (pg 14, line 278 to pg 15, line 285)

For background re access to available services please see pg 5, line 88-91

EDITORS COMMENTS: 

A rebuttal letter that responds to each point raised by the academic editor and reviewer(s). This letter should be uploaded as separate file and labelled 'Response to Reviewers'. Completed

A marked-up copy of your manuscript that highlights changes made to the original version. This file should be uploaded as separate file and labelled 'Revised Manuscript with Track Changes'. Completed

An unmarked version of your revised paper without tracked changes. This file should be uploaded as separate file and labelled 'Manuscript'. Completed

1. When submitting your revision, we need you to address these additional requirements. Please ensure that your manuscript meets PLOS ONE's style requirements, including those for file naming. The PLOS ONE style templates can be found at http://www.journals.plos.org/plosone/s/file? id=wjVg/PLOSOne_formatting_sample_main_body.pdf and http://www.journals.plos.org/plosone/s/file? id=ba62/PLOSOne_formatting_sample_title_authors_affiliations.pdf Checked and amended where required

2. In ethics statement in the manuscript and in the online submission form, please provide additional information about the patient records used in your retrospective study. Specifically, please ensure that you have discussed whether all data were fully anonymized before you accessed them and/or whether the IRB or ethics committee waived the requirement for informed consent. If patients provided informed written consent to have data from their medical records used in research, please include this information. 

Ethics statement amended (Pg 6, line 121-124) 

3. In your Methods section, please provide additional information about the methods used in your study. Please ensure you have provided sufficient details to replicate the analyses such as a description of any inclusion/exclusion criteria that were applied to participant inclusion (for example, please specify the definition of "regional northern Tasmania", and a definition of the variables used (for example, please describe how "discharge diagnosis" was coded). 

Methods section amended to address these issues.

Variables defined (pg 6, line 113-119)

Explanation of D/C diag (pg 6, line 118-119).

Northern Tasmania defined (pg 4, line 72 – 79 and pg 7, line 126 - 128)

Regional city (Launceston) defined (pg 5, line 136 – 140)

4. Please include your tables as part of your main manuscript and remove the individual files. Please note that supplementary tables (should remain/ be uploaded) as separate "supporting information" files. 

Tables included with main manuscript. 

5. We note that you have indicated that data from this study are available upon request. PLOS only allows data to be available upon request if there are legal or ethical restrictions on sharing data publicly. For information on unacceptable data access restrictions, please see http://journals.plos.org/plosone/s/data-availability#loc-unacceptable-data-accessrestrictions. In your revised cover letter, please address the following prompts: 

b) If there are no restrictions, please upload the minimal anonymized data set necessary to replicate your study findings as either Supporting Information files or to a stable, public repository and provide us with the relevant URLs, DOIs, or accession numbers. Please see http://www.bmj.com/content/340/bmj.c181.long for guidelines on how to de-identify and prepare clinical data for publication. For a list of acceptable repositories, please see http://journals.plos.org/plosone/s/data-availability#loc-recommended-repositories. We will update your Data Availability statement on your behalf to reflect the information you provide 

Thank-you for querying this.

Indication that the data from this study are available upon request was incorrect. The data provided is owned by a third party (DHHS), permission to share the data was not provided. 

All ABS data is readily available online, references are provided in the reference list along with the URL.

Thank-you for the opportunity to amend this paper.

---

## [Editor Report · Decision Letter 1]

24 Mar 2020

Socioeconomic disadvantage as a driver of non-urgent emergency department presentations: A retrospective data analysis

PONE-D-19-35242R1

Dear Dr. Unwin,

We are pleased to inform you that your manuscript has been judged scientifically suitable for publication and will be formally accepted for publication once it complies with all outstanding technical requirements.

With kind regards,

M Barton Laws

Academic Editor

PLOS ONE

Additional Editor Comments (optional):

I believe you have adequately responded to the reviewers' comments.
---

## [Editor Report · Acceptance letter]

30 Mar 2020

PONE-D-19-35242R1 

Socioeconomic disadvantage as a driver of non-urgent emergency department presentations: A retrospective data analysis 

Dear Dr. Unwin:

I am pleased to inform you that your manuscript has been deemed suitable for publication in PLOS ONE. Congratulations! Your manuscript is now with our production department. 

With kind regards,

on behalf of

Dr. M Barton Laws 

Academic Editor

PLOS ONE